# Effects of Saponins on Lipid Metabolism: The Gut–Liver Axis Plays a Key Role

**DOI:** 10.3390/nu16101514

**Published:** 2024-05-17

**Authors:** Shixi Cao, Mengqi Liu, Yao Han, Shouren Li, Xiaoyan Zhu, Defeng Li, Yinghua Shi, Boshuai Liu

**Affiliations:** 1College of Animal Science and Technology, Henan Agricultural University, Zhengzhou 450046, China; caoshixi2022@163.com (S.C.); 2019110376@sdau.edu.cn (M.L.); hy0005209056@163.com (Y.H.); lsraaa577@163.com (S.L.); zxy_0512@163.com (X.Z.); leadephone@126.com (D.L.); 2Henan Provincial Key Laboratory of Forage Resource Innovation and Utilization, Zhengzhou 450046, China; 3Henan Forage Engineering Technology Research Center, Zhengzhou 450046, China

**Keywords:** saponins, lipid metabolism disorder, gut–liver axis, lipid absorption, lipid synthesis

## Abstract

Unhealthy lifestyles (high-fat diet, smoking, alcohol consumption, too little exercise, etc.) in the current society are prone to cause lipid metabolism disorders affecting the health of the organism and inducing the occurrence of diseases. Saponins, as biologically active substances present in plants, have lipid-lowering, inflammation-reducing, and anti-atherosclerotic effects. Saponins are thought to be involved in the regulation of lipid metabolism in the body; it suppresses the appetite and, thus, reduces energy intake by modulating pro-opiomelanocortin/Cocaine amphetamine regulated transcript (POMC/CART) neurons and neuropeptide Y/agouti-related peptide (NPY/AGRP) neurons in the hypothalamus, the appetite control center. Saponins directly activate the AMP-activated protein kinase (AMPK) signaling pathway and related transcriptional regulators such as peroxisome-proliferator-activated-receptors (PPAR), CCAAT/enhancer-binding proteins (C/EBP), and sterol-regulatory element binding proteins (SREBP) increase fatty acid oxidation and inhibit lipid synthesis. It also modulates gut–liver interactions to improve lipid metabolism by regulating gut microbes and their metabolites and derivatives—short-chain fatty acids (SCFAs), bile acids (BAs), trimethylamine (TMA), lipopolysaccharide (LPS), et al. This paper reviews the positive effects of different saponins on lipid metabolism disorders, suggesting that the gut–liver axis plays a crucial role in improving lipid metabolism processes and may be used as a therapeutic target to provide new strategies for treating lipid metabolism disorders.

## 1. Introduction

Lipid metabolism disorder (LMD) is an abnormal change in the lipid profile of the blood, liver, and other tissues, including low-density lipoprotein hypercholesterolemia, hypertriglyceridemia, mixed hyperlipoproteinemia, and lowered high-density lipoprotein cholesterol alone, among other conditions [1]. Due to the fast-paced life in modern society, unhealthy lifestyles (high-fat diet, smoking, alcohol consumption, too little exercise, etc.) are more prone to LMD, where excess fat is deposited in the blood vessels and liver, which, in turn, leads to atherosclerosis, coronary artery disease, fatty liver, and other related metabolic diseases [2]. High blood lipid levels also adversely affect the gut microbiota, reducing its abundance and diversity, decreasing the relative abundance of probiotics, and increasing the abundance ratio of *Firmicutes*/*Bacteroidetes*, which leads to gut dysbiosis and further exacerbates the abnormalities of lipid metabolism [3]. The current response to disorders of lipid metabolism is mainly to reduce the risk of cardiovascular disease by lowering the concentration of low-density lipoprotein cholesterol (LDL-C) through the use of statins [4]. However, because their side effects can cause some harm to the human body (damage to the liver, digestive tract, and other functions), there is an urgent need to develop new therapeutic measures with fewer or no side effects, and, thus, naturally extracted biologically active ingredients are receiving more and more attention. For example, saponins, polyphenols, and flavones have been shown to reduce blood lipids to improve lipid metabolism disorders, and gypenosides hydrolyzed by gut microbiota were able to improve elevated blood lipids and lipid metabolism induced by high-fat diet feeding in rats [5]. Dietary polyphenols can play the role of prebiotics, increase the abundance of beneficial bacteria in the gut microbiota, regulate the production of metabolites, and alleviate lipid metabolism disorders [6]. Flavonoids extracted from Passiflora incarnata L leaf likewise produce similar effects [7].

Saponins, as a kind of bioactive components widely found in plants in nature, have been shown to have hypolipidemic, inflammation-reducing, and anti-atherosclerotic effects [8], reducing the morbidity and mortality of cardiovascular diseases, and can be used as plant extracts for the improvement of LMD. Saponin intervention in LMD is mainly achieved by regulating gut function. In addition to the direct regulation of lipid metabolism, LMD can be alleviated by restoring the composition of the gut microbiota, improving the function of the intestinal barrier, or regulating the metabolism of the microbiota, and the regulation of the gut microbiota may be an essential mechanism by which saponin intervenes in LMD. This paper reviews the health benefits of different types of saponins to improve LMD and their mechanisms of action directly or through the gut–liver axis.

## 2. Saponin

### 2.1. Sources of Saponins

Saponins, as a wide range of bioactive components within higher plants, have been found in more than 100 families of plants and small amounts in marine organisms such as starfish and sea cucumbers (Figure 1). Saponins, as plant-derived secondary metabolites, can be categorized into triterpenoid saponins and steroidal saponins, with steroidal saponins found almost exclusively in monocotyledonous plants and triterpenoid saponins found mainly in dicotyledonous plants [9]. Regular food sources of saponins include legumes, oats, tea, and Chinese yam, and non-regular food sources include ginseng, alfalfa, *Panax notoginseng*, *Bupleurum*, and *Saponaria* are mainly used for health care and as industrial raw materials [10]. Rarely does a saponin exist independently in a plant; a plant usually contains a mixture of several saponins; for example, it is known that the saponins in soybeans can be categorized into four main groups, which are group A, B, E, and DDMP [11]; twenty-four saponins were isolated and identified from alfalfa roots as 13 medicagenic acid, 2 zanhic acid, 4 hederagenin, 1 soyasapogenol A, 2 soyasapogenol B, 1 soyasapogenol C, and 1 bayogenin glycoside [12].

### 2.2. Structure and Physicochemical Properties of Saponins

Saponins, as naturally occurring glycosides, are composed of both hydrophilic sugar chain and lipophilic glycoside parts, with standard sugar chain parts including D-Glucose, L-Rhamnose, D-Glucuronic Acid, D-Xylose, L-Arabinose, and D-Galactose, and the glycoside part includes two main types of triterpenoids and sterols [13]. Most of the triterpenoid saponins are acidic, called acidic saponins, divided into tetracyclic triterpenes (lanolinanes, dammaranes, cycloartanes, cucurbitanes, etc.) and pentacyclic triterpenes (oleananes, ursanes, lupanes, friedelanes, etc.); steroidal saponins are primarily neutral, divided into spirostanol, isospirostanol, furostanol, pseudo-spirostanol, etc. (Figure 2).

Due to the amphiphilic nature of the saponin itself, it is generally an amorphous white powder, but also exists as hederagenin, which is similar to needle-like crystals, primarily bitter, hygroscopic, a natural surfactant, and mixes with water to produce a stable foam that is commonly used in cleaning products [10]. For example, *Quillaja* saponin from Chilean natives has been used for years by the indigenous people of Chile to clean their hair and clothes due to its foaming properties [14]. It has been shown to have vigorous cleaning and antimicrobial properties and a high safety profile [15]. Saponins generally have a large polarity and are soluble in water, alcohols, and aqueous alcohols, and difficult to dissolve in acetone and ether, but some reports show that some saponins can be soluble in ether solvents [16].

In addition, saponins have many pharmacological properties (Table 1) such as anti-inflammatory, antifungal, antitumor, antiviral, antioxidant, antiaging, hypoglycemic, lipid-metabolism-regulating, and organ-protective effects [10,17,18]. The combination of ginsenoside Rg3 with nanoparticles Fe_3_O_4_ was able to significantly inhibit the development and metastasis of dimethyl-nitrosamine-induced hepatocellular carcinoma and prolong the survival time of mice by remodeling the gut microbiota balance [19]. The total saponins of *Aralia taibaiensis* were able to significantly improve the oxidation-related indices in vivo in D-galactose-induced senescent rats, reversed the damages caused by D-galactose to the brain, heart, lung, kidney, liver, and spleen to different degrees, and restored the body weight, daily living ability, learning and memory ability, and organ indices of the rats [17]. Saponins are mainly used to reduce blood glucose by promoting insulin secretion, improving insulin resistance, etc. Ginsenoside Rk3 reduced serum insulin levels to promote insulin secretion through the AMPK/Akt signaling pathway, improved glucose tolerance and insulin resistance, and significantly lowered blood glucose levels in diabetic mice induced by a high-fat diet/streptozotocin [20]. This paper reviews the regulation of lipid metabolism by saponins, mainly through the inhibition of appetite, lipase activity, and fat synthesis, and the improvement of gut microbiota, and through the gut–liver axis.

### 2.3. Toxicity and Safety of Saponins

Saponins are hemolytic and can form complexes with sterols of the erythrocyte membrane, causing the rupture of the erythrocyte, resulting in an increase in permeability, hemoglobin loss, and hemolysis, and the hemolytic activity is highly correlated with its structure (sugar chains, and glycosides) [25]. Not all saponins are hemolytic; for example, ginsenosides are not hemolytic, but, after isolation, b-type, and c-type ginsenosides are significantly hemolytic, whereas a-type saponins are anti-hemolytic. Excessive use of saponins can also cause damage to metabolic organs such as the liver and kidneys. Saikosaponin A and D have been shown to cause mitochondrial apoptosis in hepatocytes leading to hepatotoxicity and liver damage [26,27]. Both the in vitro and in vivo studies showed that the spirostanol saponin terrestrosin D had potential hepatorenal toxicity. Nonetheless, hepatorenal toxicity induced by oral treatment with terrestrosin D at a dosage range of 5–15 mg/kg daily for 28 consecutive days to SD rats was reversible after 14 days of terrestrosin D withdrawal [28]. Gavage of 16 mg/kg saikosaponin D to C57BL/6J mice for 14 consecutive days inhibits the GSK3β/β-catenin pathway, suppresses cell proliferation and adult neurogenesis, and leads to impaired cognitive performance [29]. In addition, ginsenoside Rc and Re have some embryotoxicity [30]. Despite the toxicity of saponins, the amount of saponins consumed in the normal diet is usually low, and the consumption of moderate amounts of plant foods does not cause significant toxic reactions. In conclusion, saponins are toxic at certain concentrations, and ingestion at high concentrations may produce adverse effects on the organism. Therefore, plants or drugs containing high concentrations of saponins should be used with caution, and the dosage and usage of saponins should be strictly controlled to avoid these hazards.

## 3. Saponins Inhibit Fat Accumulation

### 3.1. Saponins Regulate Appetite

Appetite is regulated by hormones secreted by tissues and organs such as the hypothalamus, gastrointestinal tract, pancreas, adipose, etc. These hormones are secreted by different cells with different effects (appetite promotion and appetite suppression), which work together to regulate food intake and maintain energy balance in the system. The hypothalamus, as one of the critical control centers in the human body, plays a crucial role in regulating appetite. There are two functionally different groups of neurons in the arcuate nucleus of the hypothalamus, namely, the appetite-suppressing pro-opiomelanocortin (POMC)/cocaine amphetamine-regulated transcript (CART) neurons and the appetite-promoting neuropeptide Y (NPY)/agouti-related peptide (AGRP) neurons. POMC neurons produce appetite suppression by activating anorexigenic melanocortin 4-receptors (MC4Rs) through the release of α-melanocyte stimulating hormone (α-MSH). NPY/AGRP neurons promote appetite by releasing NPY, the MC4Rs inhibitor AGRP, and the inhibitory neurotransmitter gamma-aminobutyric acid (GABA) [31]. Ghrelin, secreted by the stomach, combines with growth hormone secretagogue receptor (GHSR) to stimulate the hypophysis to secrete growth hormone, which regulates gastrointestinal motility and, thus, increases appetite [32]. Leptin is mainly produced by white adipose tissue and binds to its receptor obesity receptor b (OBRb), which activates signal transducer and activator of transcription 3 (STAT3) phosphorylation via Janus kinase-2 (JAK2), regulating the expression of POMC and AGRP genes in the hypothalamus, stimulating POMC neurons and inhibiting NPY/AGRP neurons for appetite suppression [33]. Insulin produced by the pancreas can affect appetite by regulating blood glucose levels and peripheral cell metabolic status and can also cross the blood–brain barrier to inhibit the expression of NPY/AGRP and increase the expression of POMC/CART to achieve the effect of appetite suppression, which is similar to the role of leptin in the hypothalamus [34]. In addition, hormones such as cholecystokinin (CCK), peptide YY (PYY), and glucagon-like peptide-1 (GLP-1) have the same effect of increasing satiety and suppressing appetite [35]. Both gastrointestinal and peripheral signaling pathways ultimately act on NPY/AgRP neurons and POMC/CART neurons in the arcuate nucleus of the hypothalamus to regulate appetite through the activation or inhibition of the two, and the multiple factors interact and control each other to form an appetite-regulating network, which jointly regulates appetite.

The effects of saponins on appetite are mainly realized through regulating the appetite center, hormone secretion, and gastrointestinal peristalsis (Figure 3). Shin et al. treated the high-fat diet-induced obesity model mice with Lactobacillus plantarum-fermented *Panax notoginseng* (containing ginsenosides Rg1, Rg2, Rb1, Rd, and Rg3), which was able to significantly attenuate the expression of the NPY gene in the hypothalamus of the mice, and significantly reduce appetite to decrease energy intake [36]. Crude saponins extracted from Korean red ginseng were able to reduce serum leptin levels and hypothalamic NPY expression and decrease appetite in high-fat diet-induced obese rats, and the decrease in leptin levels may be related to the reduction of leptin-secreting adipose tissue [37]. It has been shown that protopanaxadiol- and protopanaxatriol-type saponins are effective in decreasing body weight, food consumption, and fat storage in rats by increasing the expression of CCK [38]. Astragaloside IV was able to increase leptin receptor mRNA expression in the hypothalamus of mice fed a high-fat diet in order to increase hypothalamic sensitivity to leptin, improve leptin resistance, and suppress appetite to reduce body weight [39]. In addition, saponins are involved in appetite regulation through other pathways; for example, chakasaponins from the flower buds of *Camellia* enhance the release of 5-hydroxytryptamine (5-HT), an inhibitory neurotransmitter that modulates satiety and suppresses appetite, in the isolated ileum of mice, and chakasaponins similarly inhibited the expression of hypothalamic NPY mRNA [40].

### 3.2. Saponins Inhibit Pancreatic Lipase Activity

Lipase, as one of the main digestive enzymes, can hydrolyze dietary fats into glycerol and fatty acids for the next step of absorption, and it is mainly pancreatic lipase that plays a role in the gastrointestinal tract, so the inhibition of pancreatic lipase activity can be used as one of the means to reduce lipid absorption. Orlistat, as an effective inhibitor of pancreatic lipase that reduces the absorption of fat by the organism, has been approved for treating obesity. However, its safety and long-term stability are yet to be determined due to its susceptibility to gastrointestinal adverse effects (diarrhea, abdominal pain, oily feces, etc.) [41]. As a natural bioactive ingredient, saponin has been widely reported as a potential natural inhibitor of pancreatic lipase [42], which inhibits lipase activity and suppresses the process of fat digestion and absorption to alleviate obesity through mechanisms such as competition or noncompetition. It has been shown that saponin extracts and their hydrolyzed products from fenugreek and quinoa seeds significantly inhibited porcine pancreatic lipase activity in an in vitro digestion model and that the half-maximal inhibitory concentration (IC50) was significantly correlated with the saponin content of the extracts [43]. Yoshizumi et al. found that sessiloside and chiisanoside saponins extracted from the leaves of *Acanthopanax sessiliflorus* were able to inhibit porcine pancreatic lipase activity in vitro to varying degrees, exhibiting dose-dependent IC50 values of 0.36 and 0.75 mg/mL, respectively [44]. Most species of saponins exhibit varying degrees of pancreatic lipase inhibition, and the inhibitory effects of different species of saponins on lipase vary, with parameters such as inhibition rates and IC50 values altered by different substituent groups, number of rings, and chemical structures. In conclusion, saponins indirectly improve lipid metabolism in obese patients by reducing lipid absorption and energy intake due to their property of inhibiting pancreatic lipase activity.

### 3.3. Saponins Regulate the Synthesis and Breakdown of Fats

The accumulation of fat, as the primary energy storage material of the organism, consists of an increase in the volume and number of adipocytes. The increase in volume is mainly because the energy intake of the organism is higher than the energy consumed. The excess energy is converted into triglycerides and stored in adipocytes, and the increase in number refers to the formation of mature adipocytes through the differentiation of adipose precursor cells. The process is mainly influenced by the AMP-activated protein kinase (AMPK) signaling pathway and related transcriptional regulators such as peroxisome-proliferator-activated-receptors (PPAR), CCAAT/enhancer-binding proteins (C/EBP), sterol-regulatory element binding proteins (SREBP), and adipokines such as adiponectin [45,46,47]. The inhibitory effect of saponins on fat synthesis is also mainly realized through these pathways (Figure 4).

AMPK, as a regulatory center of lipid metabolism, can be activated by sensing a decrease in organic energy, which, in turn, increases acetyl CoA carboxylase (ACC) phosphorylation, decreases ACC activity, reduces the conversion of acetyl CoA to malonyl CoA, and enhances the activity of carnitine palmitoyl transferase-1 (CPT-1), which promotes the oxidation of fatty acids for energy production and, ultimately, reduces fat synthesis [48]. AMPK also activates adipose triglyceride lipase (ATGL) and hormone-sensitive triglyceride lipase (HSL) to promote lipolysis [49]. It has been shown that kinsenoside can activate AMPK in C3H10T1/2 adipocytes, significantly increase the expression of ATGL and CPT-1 proteins, and promote lipolysis and metabolism [50].

PPAR is involved in fatty acid metabolism, adipocyte differentiation, and other processes regulating various target genes; PPARs include three isoforms: PPAR-α, PPAR-β, and PPAR-γ. PPAR-α is mainly expressed in tissues that can catabolize fatty acids, such as liver and skeletal muscle. PPAR-β is widely distributed throughout the body, and PPAR-γ is mainly expressed in white and brown adipose tissues, with differences in the distribution of the isoforms giving specific actions to the regulators of PPARs in different tissues [51]. Saponin, as a natural potential regulator of PPARs, can participate in processes such as fatty acid oxidation, as well as adipocyte differentiation by modulating PPARs [52]. For example, gypenosides significantly upregulated PPAR-α mRNA and protein levels in the serum and liver of rats with fatty liver [53], and oleuropein regulated lipid metabolism through hte activation of PPAR-α both in vivo and in vitro [54]. Li et al. treated *db*/*db* mice by gavage with gymnemic acid, significantly increased the expression of PPAR-β, and enhanced fatty acid oxidation to reduce lipid accumulation in the liver and skeletal muscle of mice [55]. It has been shown that ginsenoside Rg2 significantly reduced PPAR-γ mRNA and protein expression levels in high-fat diet-induced obese mice and 3T3-L1 to inhibit adipocyte differentiation and adipogenesis [56], and that soyasaponins Aa and Ab likewise had similar effects [57]. In addition, ginsenosides Rg3 and Rb1 increased the expression of browning-related genes, such as brown adipose-specific uncoupling protein 1 (UCP1), by increasing the phosphorylation of AMPK and regulating the expression of PPAR-γ in 3T3-L1 cells, respectively, and it induces the browning of white adipocytes and reduces the deposition of lipid droplets in the cells, which is an anti-obesity effect [58,59].

C/EBP is one of the essential transcription factors regulating the growth and differentiation of adipocytes and can regulate adipogenesis together with a variety of cytokines and transcription factors, with a total of six subtypes, of which C/EBP-α, C/EBP-β, and C/EBP-δ are highly expressed in adipocytes [60]. It was shown that bioactive components (including saponins, with the dried leaves containing higher saponins than the green leaves) extracted from the green and dried leaves of Korean ginseng were able to significantly reduce the levels of mRNA and protein expression of C/EBP-α and PPAR-γ, and their downstream target genes, such as lipocalin, in the adipose tissue of the epididymis of high-fat diet-fed rats. In addition, the dried leaf extract was able to significantly reduce the mRNA of C/EBPs (α, β, and δ), PPAR-γ, and adiponectin, as well as the protein expression levels of C/EBP-α, PPAR-γ, and adiponectin, inhibit adipogenesis, and reduce lipid accumulation in 3T3-L1 adipocytes [61]. Similarly, ginsenoside Rh1 was able to significantly reduce the mRNA and protein expression of C/EBP-α and PPAR-γ in high-fat diet-induced obese mice, and 3T3-L1 cells have arrived at the inhibition of adipocyte differentiation and reduction of fat-hoarding [62].

SREBP-1 mainly regulates genes related to fatty acid and cholesterol synthesis, such as ACC; SREBP-2 mainly controls the transcription of cholesterol synthesis genes, such as 3-hydroxy-3-methylglutaryl-CoA synthase (HMGCS). In adult liver and adipocytes, SREBP-1 synthesizes fatty acids and triglycerides, mainly in the form of SREBP-1c. AMPK phosphorylation inhibits SREBP-1c expression, reduces fatty acid synthetase (FAS) mRNA levels, enhances fatty acid oxidation, reduces lipid synthesis, and improves lipid metabolism disorders [63]. Diosgenin was able to upregulate the expression of AMPK, ACC, and CPT-1 in LO2 cells to increase lipolysis and inhibit the expression of lipid-synthesis-related proteins SREBP-1c and FAS through the AMPK-ACC-CPT-1 signaling pathway to reduce lipid accumulation [64]. In addition, Saikosaponin D upregulated the expression of insulin-inducible genes 1/2 (INSIG1/2), inhibited SREBP-1c activity, promoted fatty acid oxidation, and inhibited fatty acid synthesis through the activation of PPAR-α in hepatocytes and adipocytes to ameliorate the accumulation of lipids in the liver and adipocytes due to the feeding of a high-fat, high-glucose diet in mice [65]. Soybean saponin was likewise able to reduce mice hepatic SREBP-1c and FAS mRNA levels [66]. Gymnemic acids, on the other hand, reduce lipid accumulation in HepG2 cells by inhibiting liver X receptor (LXR), downregulating its activation of SREBP-1c [67].

## 4. Saponins Regulate Lipid Metabolism via the Gut–Liver Axis

With the development of biotechnology, especially the rise of the microbiome, increasingly more studies have shown that the gut microbiota is closely linked to the organs and tissues of the animal body, and this association is called an “axis”, such as the gut–brain axis and the gut–liver axis. The gut–liver axis refers to the relationship between the interaction of the gut and the microbes and metabolites in it and the liver and the bile it secretes, and is used to describe the biological link between the gut and the liver, which is the result of the interaction of factors such as diet, genetics, and the environment. The portal vein, as the connector between the intestine and the liver, bears the burden of maintaining blood circulation between the intestine and the liver; about two-thirds of the liver’s blood comes from the intestine, and intestinal derivatives can reach the liver through the portal vein, and bile acids produced by the liver influence the intestine and the microbiota therein through it [68]. The gut microbiota affects the liver as well as lipid metabolism through different pathways, mainly through their metabolites or derivatives such as short-chain fatty acids (SCFAs), bile acids (BAs), trimethylamine (TMA), and lipopolysaccharide (LPS) [69].

### 4.1. Protective Effects of Saponins on the Gut–Liver Axis

#### 4.1.1. Protective Effects of Saponins on the Intestinal Barrier

The intestinal barrier plays a vital role in the interaction between the gut and the liver as the basis for stabilizing the gut–liver axis, and a sound intestinal barrier protects the liver from harmful gut microbiota and harmful substances through the portal vein. The intestinal barrier consists of a microbial barrier (gut microbiota), a chemical barrier (mucus, digestive fluids, and microbial secretions), a mechanical barrier (mucosal epithelium, lamina propria, and muscularis mucosa), and an immune barrier (lymphocyte tissue and immune cells). Once the various barriers of the intestinal tract are disrupted and functionally impaired, all types of antigens, harmful substances, and pathogenic micro-organisms in the intestinal tract enter the bloodstream via the portal vein, which, in addition to generating liver damage, also leads to damage to other systems and organs [70]. High-fat diet, obesity, and other factors make the healthy intestinal microecology disordered; the intestinal barrier function is impaired, and the imbalance of microbiota and their metabolites enter the liver through the damaged intestinal barrier and portal vein, which leads to the release of inflammatory factors through the activation of hepatic Toll-like receptors, causing metabolic disorders, such as insulin resistance, as well as liver disease [71]. Saponins, as a widely available bioactive substance, have been shown to enhance intestinal barrier function by reducing tissue damage, maintaining the mucus barrier integrity, reducing inflammatory responses, and altering intestinal permeability [72]. It has been shown that dioscin can inhibit the expression of inflammatory pathways to alleviate intestinal barrier damage in mice by decreasing the secretion of colonic pro-inflammatory factors, increasing anti-inflammatory factors, and enhancing the expression of intestinal tight junction proteins [73]. Protopanaxatriol-type saponin was able to ameliorate the dysbiosis of the gut microbiota in mice caused by antibiotic infestation, increase the levels of metabolites such as SCFAs, improve intestinal permeability, and maintain intestinal internal environmental homeostasis [74]. In addition, Ilexhainanoside D and ilexsaponin A have been shown to restore intestinal barrier damage induced by a high-fat diet in mice by modulating the gut microbiota, as well as increasing the intestinal tight junction protein expression [75]. In conclusion, saponins can positively influence intestinal barrier function and contribute to maintaining gut–liver axis homeostasis.

#### 4.1.2. Protective Effects of Saponins on the Liver

As the largest digestive gland in the organism and a vital organ involved in metabolism, the liver plays a huge role in regulating lipid metabolism. Ginsenoside CK, as a product of ginsenoside Rb1 and other natural ginsenosides metabolized in the intestine, has a variety of pharmacological activities, and Zhang et al. found that ginsenoside CK significantly ameliorated fructose-induced hepatic steatosis and oxidative damage, alleviated hepatic inflammation, and activated the AMPK signaling pathway to regulate hepatic lipid metabolism disorders in mice [76]. Saponins extracted from cherry tomatoes likewise ameliorated liver injury in mice induced by high-fat diets and reduced serum levels of liver injury markers (ALT, AST, etc.) [77]. A study showed that *Panax ginseng* saponins were able to positively affect ethanol-induced alcoholic fatty liver mice by decreasing hepatic endogenous oxidants (ROS, and MDA), increasing the content of antioxidants (GSH, and SOD), and decreasing the levels of inflammatory factors (IL-6, and TNF-α) in order to reduce the oxidative damage and inflammatory response in the liver induced by ethanol treatment [78]. In addition, ginsenoside Rg1 reduces hepatic reactive oxygen species and pro-inflammatory factors through NF-κB and ameliorates aging-induced hepatic injury such as hepatic fibrosis, inflammation, and oxidative stress [79]. This shows that saponins can protect the liver and reduce the damage it receives when external factors affect it.

### 4.2. Saponins Regulate Lipid Metabolism via the Gut–Liver Axis

Saponins regulate lipid metabolism by the gut–liver axis primarily through the modulation of the gut microbial composition and the metabolites they produce (Figure 5).

#### 4.2.1. Short-Chain Fatty Acids

SCFAs are products of the fermentation of indigestible dietary polysaccharides, especially dietary fiber, by the microbiota in the posterior portion of the intestine and consist mainly of acetic acid, propionic acid, and butyric acid. Different gut microbiota produces corresponding different types of SCFAs, with the Firmicutes (e.g., *Ruminalococcaceae*) producing mainly butyric acid and the Bacteroidetes (e.g., *Bifidobacterium adolescentis* and *Prevotell*) producing mainly acetic acid and propionic acid [80]. SCFAs are mainly butyric acid that can be absorbed by colonic epithelial cells and converted to acetyl coenzyme A after β-oxidation to participate in the TCA cycle for cellular energy supply, and, after metabolism, they can provide approximately 70% and 10% of the required calories for grass-feeding ruminants, and omnivorous animals, such as humans and pigs, respectively [81]; propionic acid reaches the liver via the portal vein and is metabolized by hepatocytes, while acetic acid from co-transportation remains in the liver or is released systemically via peripheral veins [82]. SCFAs can regulate metabolism and reduce inflammation through histone deacetylase (HDAC) and three types of G Protein-Coupled Receptors (GPCRs)—GPR41, GPR43, and GPR109A. GPR41 and GPR43 can also be called free fatty acid receptor 3 (FFAR3) and free fatty acid receptor 2 (FFAR2), respectively. GPR43 is mainly distributed in intestinal epithelial cells, enteroendocrine cells, pancreatic β-cells, immune cells, and white adipocytes, and is involved in the regulation of inflammation and glycolipid metabolism, as well as regulating the functional expression of PYY and GLP-1; GPR41 is mainly distributed in peripheral neuronal cells, enteroendocrine cells, pancreatic β-cells, immune cells, and white adipocytes, and is involved in the regulation of lipid distribution [83]. GPR41-/- and GPR43-/- mice have a weaker immunity to respond to inflammation and infection induced by ethanol, 2,4,6 trinitrobenzene sulfonic acid (TNBS), or *Citrobacter rodentium*, as well as a slower clearance of bacteria compared to normal mice, and GPR41 and GPR43 regulate the production of chemokines and cytokines in epithelial cells through the activation of the extracellular protein kinase (ERK1/2) and p38 mitogen-activated protein kinase (MAPK) signaling pathways that regulate the immune-inflammatory process [84]. SCFAs regulate gene expression and organismal immune responses by inhibiting HDAC, and SCFAs enhance immune function in mice by inhibiting the HDCA upregulation of B10 cells and downregulation of colonic inflammatory factors in DSS-induced colitis mice [85]. SCFAs can inhibit the growth of pathogenic bacteria in the intestinal tract, inhibit intestinal inflammation, and improve the intestinal barrier by inhibiting the NF-κB signaling pathway to reduce the expression of pro-inflammatory factors such as TNF-α, IL-6, and IL-12 and increase the expression of the anti-inflammatory factor IL-10, and increase the production of antimicrobial peptide (AMP) and Treg cells [86]. Upon reaching the liver, SCFAs can activate AMPK phosphorylation, promote downstream PGC-1α expression, and regulate the transcription factor PPAR-α, which acts on the LXR and farnesoid X receptor (FXR), thereby affecting cholesterol and lipid metabolism [87]. SCFAs also reduce PPAR-γ activity and convert intrahepatic lipogenesis to fatty acid oxidation via the UCP1-AMPK-ACC signaling pathway [88]. In addition, SCFAs produced by the gut microbiota were able to increase the biosynthesis of membrane phospholipids in the livers of partially hepatectomized mice to improve liver regeneration [89].

Diammonium glycyrrhizinate (DG), a steroidal saponin from Glycyrrhiza glabra, has been used to treat liver diseases. Li et al. used DG in high-fat diet-fed mice. They found that it increased the relative abundance of SCFA-producing bacteria such as *Ruminococcaceae* and *Lachnospiraceae* and increased the content of SCFAs in the intestine to reduce body weight and improve hepatic steatosis in mice [90]. It has been shown that mogrosides were able to improve the gut microbiota of rats fed a high-fat diet, reduce the ratio of *Firmicutes*/*Bacteroidetes*, improve insulin resistance, increase the levels of acetic acid and butyric acid to activate the AMPK-related pathway, reduce the levels of serum TC, TG, and LDL-C, and improve the disorders of glucose–lipid metabolism [91]. Zhou et al. showed that Akebia saponin D increased the relative abundance of SCFA-producing bacteria *norank_f_Bacteroidales_S24-7_group*, *Ruminococcus_1*, and *Lachnospiraceae_NK4A136_group* to alleviate hyperlipidemia induced by high-fat diet feeding rats [92]. In addition, saponins such as xanthoceraside, *Pulsatilla chinensis* saponins, and forsythiaside A can modulate gut microbiota homeostasis and increase the production of SCFAs [93,94,95]. These results suggest that saponins can ameliorate lipid metabolism disorders in the body by modulating the production of SCFAs by the gut microbiota.

#### 4.2.2. Bile Acids

As one of the significant metabolites linking the gut and liver, BAs are an integral part of the gut–liver axis. Different levels of bile acids in the gut affect the composition and abundance of the gut microbiota, and, conversely, the gut microbiota affect the biotransformation of bile acids [96,97]. BAs are divided into primary bile acids (synthesized from cholesterol in the liver via the “Classical” and “Alternative” pathways) and secondary bile acids (formed from primary bile acids chemically modified by the gut microbiota); e.g., hepatic production of cholic acid (CA) and chenodeoxycholic acid (CDCA) can be converted to deoxycholic acid (DCA) and lithocholic acid (LCA) via 7α-dehydroxylation with the intervention of the gut microbiota [98]. Bile acids, as one of the main components of bile, are synthesized by the liver and stored in the gallbladder, which flows into the duodenum with bile after food intake to promote intestinal lipid absorption, and, through passive diffusion and active transport, about 95% of the bile acids are transporter-absorbed at the terminal end of the ileum, and returned to the liver through the portal vein, a process known as the enterohepatic recycling of bile acids, and the rest of the bile acids, which is about 5%, are excreted with the feces [99].

Bile acids can influence host metabolic pathways and disease progression by activating various nuclear and G-protein-coupled receptors, primarily FXR and G-protein-coupled bile acid receptor 1 (TGR5). FXR activation in the liver and gut inhibits CYP7A1 and CYP8B1 to regulate bile acid homeostasis mainly through the activation of small heterodimeric ligand (SHP) and fibroblast growth factor 15/19 (FGF15/19) and fibroblast growth factor receptor 4 [100]. In the gut, FXR can produce antimicrobial peptides that work with the antibacterial properties of bile acids to inhibit bacterial overgrowth and regulate gut microbiota homeostasis. The inhibition of intestinal FXR reduces the expression of various ceramide synthases to lower intestinal and blood ceramide levels, and the inhibition of ceramide biosynthesis as a lipotoxic inducer of metabolic disorders ameliorates a variety of metabolic diseases [101]. Cen Xie et al. were able to selectively inhibit intestinal FXR signaling by supplementing caffeic acid phenethyl ester (CAPE) to increase tauro-β-muricholic acid levels (an FXR inhibitor) in high-fat diet-fed mice to ameliorate glucose metabolism dysfunction and regulate hepatic glucose isomerization through the intestinal FXR–ceramide pathway [102]. The TGR5 receptor activation by bile acids can mediate critical signal transduction pathways such as nuclear factor-κB (NF-κB), protein kinase B (AKT), extracellular regulated protein kinases (ERK), and signal transducer and activator of transcription 3 (STAT3) [103]. Among them, the TGR5-AKT-mTORC and TGR5-AKT-ERK1/2 pathways are involved in regulating metabolism, alleviating insulin resistance, regulating hepatic lipid metabolism, and ameliorating metabolic disorders [104,105]. The activation of TGR5 also regulates metabolic homeostasis by promoting GLP-1 secretion from enteroendocrine L cells to improve hepatic glucose homeostasis and insulin sensitivity and increase adipose tissue browning [106]. Kavita Jadhav et al. used INT-767 to activate FXR and TGR5 receptors through their downstream signaling pathways to achieve the inhibition of adipogenesis to ameliorate metabolic disorders, suggesting that FXR and TGR5 play an essential role in the regulation of metabolism in the organism [107]. LXR has two subtypes, LXR-α and LXR-β, which are regulatory sensors of lipid metabolism and cholesterol levels and regulate lipogenesis in the liver through the LXR-SREBP-1C pathway [108]. Thus, BAs are regarded as signaling molecules that act in both directions of the gut–liver axis and play essential roles in regulating gut microbiota composition, hepatic lipogenesis, inflammatory responses, and bile acid profiles. As a health food, ginseng has many effects. Ginsenoside Rg1 can improve gut microbiota disorders in mice caused by feeding high-fat diets, increase the relative abundance of *Lachnoclostridium*, *Streptococcus*, *Lactococcus*, *Enterococcus*, and *Erysipelatoclostridium*, increase the levels of CA and TCA significantly in bile acids, decrease TC and TG levels, increase UCP-1 expression in brown adipose tissue, and increase adipose thermogenesis to ameliorate obesity [109]. Diosgenin is a steroidal saponin widely found in Dioscoreaceae plants. Yan et al. showed that diosgenin could improve the gut microbiota of non-alcoholic fatty liver disease mice, increase the relative abundance of *Clostridia*, activate FXR and its downstream pathway, and regulate the metabolism of Bas, as well as the enterohepatic circulation, reduce the levels of TC, ALT, and TG, and improve the fatty liver disease [110]. Saikosaponin A, a bioactive component of the traditional Chinese medicine *Radix Bupleuri*, was shown to decrease the abundance of bile salt hydrolase-producing bacteria such as *Lactobacillus*, *Bifidobacterium*, and *Turicibacter* by modulating the gut microbiota of laying hens fed a high-energy diet, increasing the levels of Taurochenodeoxycholic acid and Tauro-α-muricholic acid levels, decreasing BA synthase gene expression levels, and inhibiting BA reabsorption, thereby increasing BA excretion and reducing cholesterol and lipid accumulation [111]. A study in The Journal of Clinical Investigation stated that alfalfa top saponins reduced total cholesterol levels in Macaques and had no significant effect on HDL-C, decreased cholesterol absorption, and increased bile acid excretion to alleviate hypercholesterolemia induced by a high-cholesterol diet [112]. It can be seen that saponin regulates the gut microbiota, activates FXR- and TGR5-related signaling pathways, regulates glucose homeostasis, improves insulin resistance, and maintains bile acid homeostasis while increasing the excretion of bile acids to lower cholesterol and regulate metabolic disorders.

#### 4.2.3. Trimethylamine

The gut microbiota trimethylamine lyase converts TMA from nutrients rich in choline, L-carnitine betaine, and phosphatidylcholine. TMA is transferred to the liver via the portal vein to be converted to trimethylamine-*N*-oxide (TMAO) by oxidation with flavin monooxygenase (FMO). Plasma TMAO levels are closely related to the gut microbiota composition, and it has been shown that some gut microbiota from the phylum *Firmicutes* and *Bacteroidetes* are positively correlated with plasma TMAO levels, including *Prevotella* and *Clostridium* [113]. TMAO promotes fat deposition by affecting cholesterol transport and inhibiting bile acid production, and TMAO activates FXR and SHP, inhibits CYP7A1 and SREBP-1C expression to inhibit reverse cholesterol transport, and inhibits bile acid synthesis, leading to lipid deposition [114]. In addition, TMAO increased forkhead box-containing protein O subfamily-1 (FoxO1) expression levels by inhibiting the phosphatidylinositol 3-kinase (PI3K)-AKT pathway and its downstream Insulin Receptor Substrate 2 (IRS2, PI3Kr1, and AKT2) genes, and blocked insulin signaling; it also inhibits the expression of Liver-Specific Glycogen Synthase (GYS2), exacerbating the impairment of hepatic glycogen synthesis; and reduces the expression level of Glucose Transporter2 (GLUT2), which reduces the hepatic glycogen transport capacity, ultimately leading to fat deposition in the body [115]. Gypenosides were able to reverse choline-induced changes in the gut microbiota of mice and reduce the levels of gut-microbiota-derived trimethylamine lyase, thereby reducing TMA and TMAO levels, and gypenosides also adjusted the expression of lipid-metabolism-related enzymes in response to choline-induced elevation of TG in mice, leading to lipid-lowering effects [116].

Ginsenosides Rh2 was able to reduce the abundance of trimethylamine-lyase-producing bacteria, such as *Clostridium*, in the gut microbiota, lowering serum TMA levels and reducing the damage caused by TMAO [117]. In addition, Saikosaponin A can reduce fat deposition by decreasing FMO activity and reducing TMAO production [118]. Thus, saponins improve lipid metabolism mainly by modulating the effect of the gut microbiota on the intermediate enzymes of the TMA conversion process, decreasing the level of TMAO in the gut–hepatic axis.

#### 4.2.4. Lipopolysaccharide

LPS, a complex of lipids and polysaccharides, is an essential component of the cell wall of Gram-negative bacteria and is released into the body after bacterial death. LPS is generally not directly involved in lipid metabolism, but, rather, by inducing intestinal inflammation, it enters the enterohepatic axis and affects the liver, leading to metabolic disturbances that exacerbate disease progression. A high-fat diet can disrupt gut microbiota homeostasis, destroy the intestinal barrier to increase intestinal permeability, increase the level of LPS in the intestinal tract, and enter the blood circulatory system through the intestinal barrier to cause endotoxemia [119]. It has been shown that feeding mice a high-fat diet increases the relative abundance of Gram-negative bacteria in their gut microbiota, decreases the levels of SCFAs, and increases the levels of LPS, causing inflammation, as well as metabolic disorders, and affecting the health of the body [120]. LPS can activate TLR4 in macrophages, hepatocytes, adipocytes, and other cells, and activate NF-κB through MyD88 to induce adipose and hepatic inflammatory responses and promote lipid accumulation [121]; in addition, the activation of the NF-κB signaling pathway increases the ROS content, causing oxidative damage in the liver [122], and enteric-derived LPS is also capable of inducing insulin resistance, leading to disorders of glucose–lipid metabolism weight gain [123]. Intestinal-derived high-density lipoprotein (HDL) reaches the liver through the portal vein in the form of small HDL HDL3, which binds to LPS from the intestines via LPS-binding protein (LBP) to form a complex that prevents it from binding to Kupffer cells and attenuates the hepatic inflammatory response brought about by LPS [124]. However, a recent study has shown that LPS-induced sustained inflammation through NF-κB can increase the levels of adipose browning markers, such as UCP-1, in the adipose tissue of subject mice, promote the transformation of white fat to brown fat, and increase adipose thermogenesis to reduce fat accumulation [125], which suggests that LPS plays a complex role in the gut–liver axis.

Ilexhainanoside D and ilexsaponin A, while restoring the high-fat diet-induced gut microbiota disruption and altered intestinal permeability, correspondingly reduced the concentration of LPS in the bloodstream and attenuated LPS-induced inflammatory responses and lipid accumulation [75]. *Panax notoginseng* saponins were able to improve the intestinal fistula phenomenon, regulate adipogenesis, fatty acid transport, and oxidative gene expression, reduce hepatic lipid degeneration, and ameliorate hepatic fibrosis and lipid metabolism disorders in high-fat diet-fed mice by inhibiting TLR4 through reducing LPS [126]. It has been shown that sarsasapogenin can ameliorate exogenous LPS-induced adipose inflammation in mice, as well as insulin resistance in high-fat diet-induced obese mice, and regulate lipid metabolism [127]. In conclusion, saponins mainly improve the intestinal barrier by regulating the gut microbiota, reducing the total amount of LPS entering the gut–liver axis, protecting the stability of the gut–liver axis, and reducing inflammation and metabolic abnormalities.

## 5. Conclusions and Outlook

Obesity-induced lipid metabolism disorder seriously affects the daily life of people; saponin, widely existing in terrestrial higher plants and some marine organisms in the bioactive ingredients, can reduce lipid absorption, inhibit fat synthesis, and, through the gut–liver axis interactions to regulate lipid metabolism, improve lipid metabolism disorders. As a naturally occurring bioactive ingredient, saponin has more targets, can activate a wider range of signaling pathways, and has fewer side effects than drugs, which proves that it has a wide range of future applications in the treatment of metabolic diseases associated with obesity. However, the function of saponins and the mechanism of their interaction with the gut–liver axis have not been fully discovered. For example, the effect of intestinal microbial changes on the hydrolysis of saponins, whether saponins can directly act on related receptors, and the specific mechanism affecting lipid metabolism are still unclear, and a large number of experiments are still needed to be explored.

## Figures and Tables

**Figure 1 nutrients-16-01514-f001:**
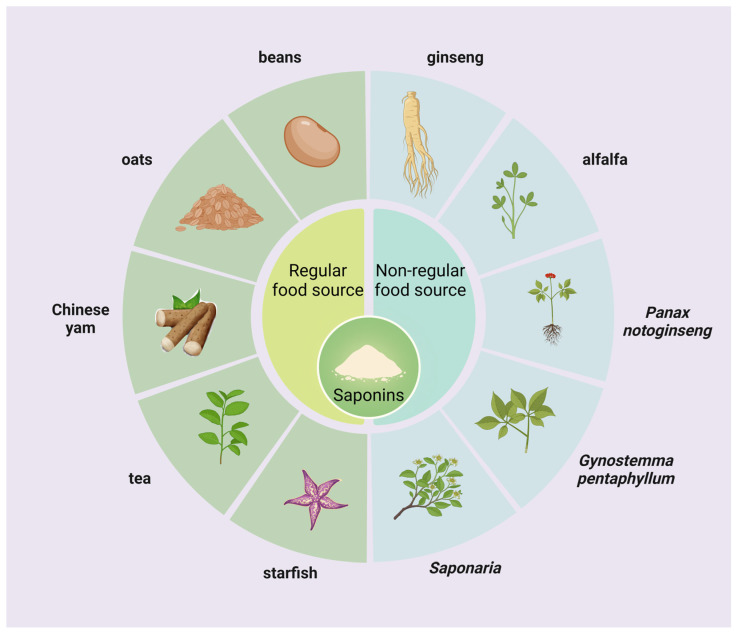
Sources of saponins. Saponins are widely found in higher land plants and some marine organisms and can be categorized into regular food and non-regular food sources.

**Figure 2 nutrients-16-01514-f002:**
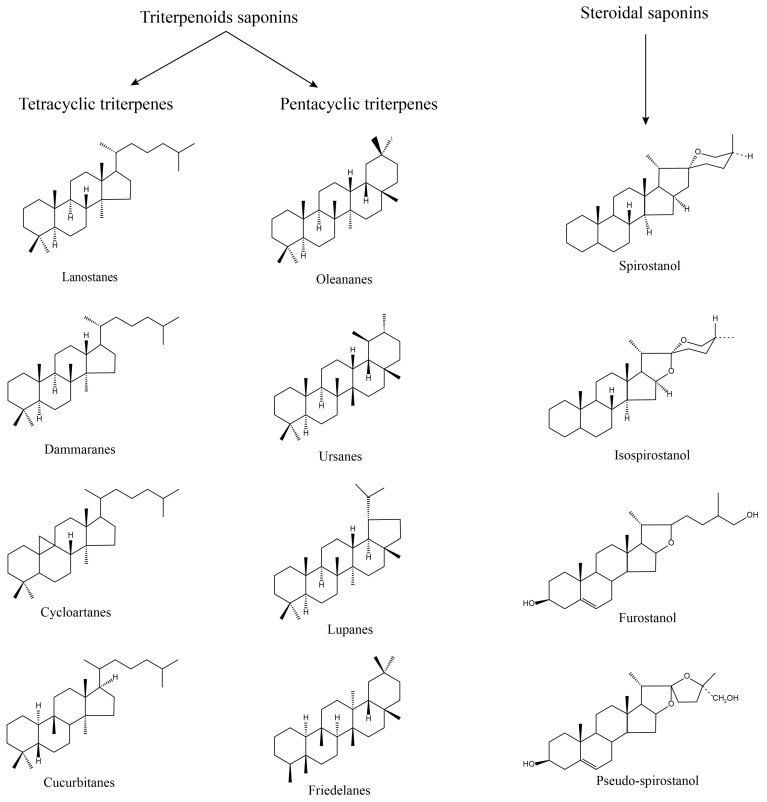
Structure of saponins. Saponin consists of two parts: sugar chain and glycoside; according to the structure, it can be divided into triterpenoids saponin and steroidal saponin, of which triterpenoids saponin can be divided into tetracyclic triterpene and pentacyclic triterpene.

**Figure 3 nutrients-16-01514-f003:**
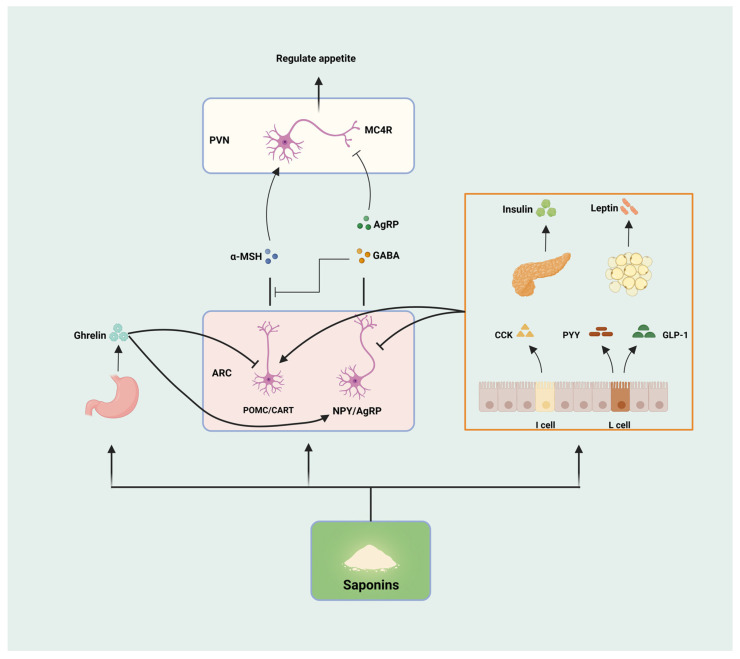
Saponins regulate appetite. Saponins can be involved in the regulation of appetite through the modulation of hypothalamic appetite centers (POMC/CART, and NPY/AgRP), hormone secretion (Ghrelin, Insulin, Leptin, CCK, PYY, and GLP-1), and gastrointestinal peristalsis, respectively.

**Figure 4 nutrients-16-01514-f004:**
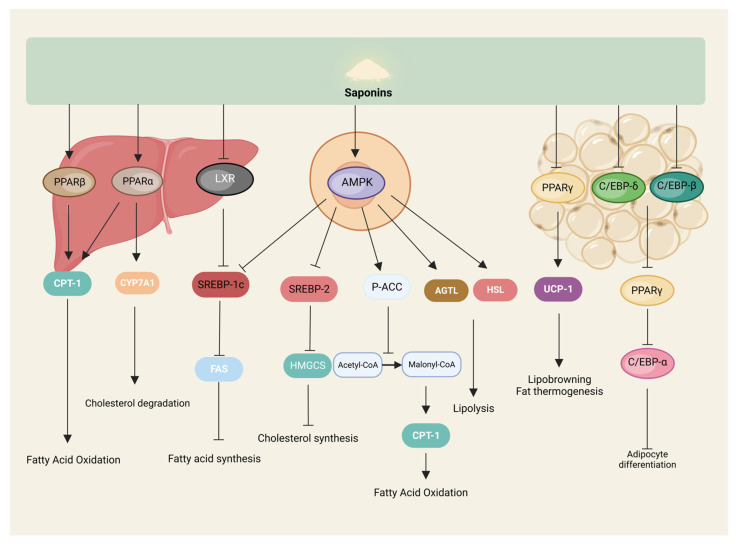
Saponins inhibit lipogenesis. Saponins modulate signaling pathways related to lipid metabolism in the liver, cells, and adipose tissue to participate in lipolysis and lipogenesis.

**Figure 5 nutrients-16-01514-f005:**
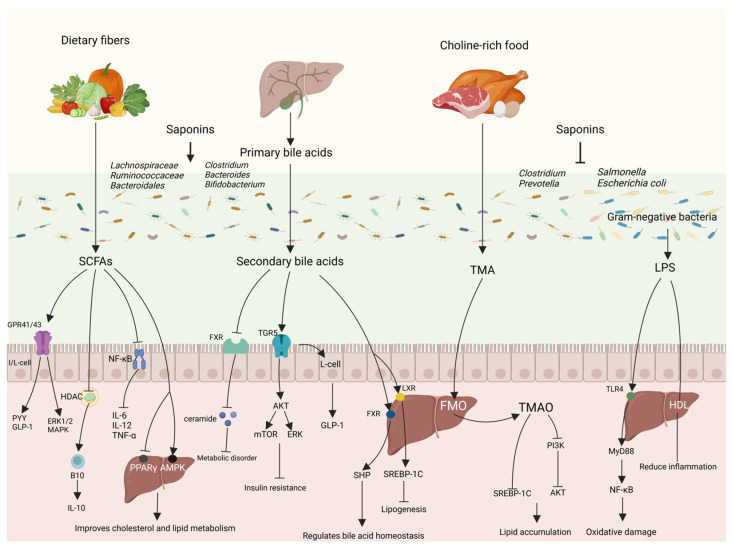
Saponins regulate lipid metabolism through gut–liver axis. Saponins are involved in regulating lipid metabolism by modulating the structure of the intestinal microbial community, altering the level of metabolites of the flora, and, thus, through intestinal–hepatic interactions.

**Table 1 nutrients-16-01514-t001:** Pharmacological properties of different saponins.

Saponin	Pharmacological Properties	Model	References
Garlic saponins	Lipid lowering, lowering of cholesterol	mice	[10]
Ginsenosides	Anticancer activity	mice	[10]
*Aralia taibaiensis* saponins	Antioxidant, anti-aging, organ protection	rat	[17]
Gnsenoside Rb2	Improvement of fatty liver disease	mice	[18]
Gnsenoside Rg3	Anticancer activity	mice	[19]
Gnsenoside Rk3	Lowering blood glucose, improving insulin resistance	mice	[20]
Sarsasapogenin	Improvement of rheumatoid arthritis	rat	[21]
Kuding tea saponins	Decrease serum lipids, improve lipid metabolism	mice	[22]
Gypenosides	Enhancement of hypoxia tolerance	mice	[23]
Diosgenin	Anticancer activity	rat	[24]

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
