# Peer review of "Effects of Saponins on Lipid Metabolism: The Gut–Liver Axis Plays a Key Role"

_nutrients, 2024, doi:10.3390/nu16101514_

Round 1
Reviewer 1 Report
Comments and Suggestions for Authors
The review addresses a relevant topic for clinical medicine and health improvement, by showing natural alternatives for improving metabolic changes associated with diseases such as obesity. Therefore, I will make considerations to improve the review.
1 - Why the authors chose to focus on saponins? The polyphenols are also natural compounds with many actions on the gut-liver axis.
2 - Page 5, line 145: change the citation Na RAe Shin to Shin et al., as in the references section.
3 - Does saponin inhibit pancreatic lipase activity in a similar way to Orlistat?
4 - Figure 5 The therm saponins are spelled wrong, it is spoanins. Please correct.
5 - Review the text as there is no space between the end of a sentence and the beginning of another sentence, or between the end and the reference number.
6 - Rewrite the first three lines of the conclusion (357,358,359), as they are meaningless
Reviewer 2 Report
Comments and Suggestions for Authors
ABSTRACT. Since you have not exceeded the limit of words in the abstract, additional results that obtained in your revision work can and should be included.
INTRODUCTION. Line 37. Besides saponins, could you provide additional examples of natural bioactive ingredients?
SOURCES OF SAPONINS. Lines 56-57. The term ‘non-food sources’ is not clear to me in this context. Why, for instance, are alfalfa and ginseng considered ‘non-food sources’?
SOURCES OF SAPONINS. Please, revise that scientific names of species are written correctly and in italics (in the main text and in Figures, not only in this section but throughout the whole manuscript).
SAPONINS ACTIVITIES. A table summarizing the revised studies is strongly recommended, compilating the effects, the responsible molecule, the experimental model, etc.
CONCLUSIONS AND OUTLOOK. Although current limitations are mentioned, future perspectives should be described more explicitly.
